# The Association between Iron and Folic Acid Supplementation and Malaria Prophylaxis and Linear Growth among Children and Neonatal Mortality in Sub-Saharan Africa—A Pooled Analysis

**DOI:** 10.3390/nu14214496

**Published:** 2022-10-26

**Authors:** Deepali Godha, Manisha Tharaney, Simeon Nanama, Tina Sanghvi, Arnaud Laillou, Fanta Touré Diop, Aita Sarr Cisse

**Affiliations:** 1Independent Researcher, Indore 452016, India; 2Nutrition, FHI 360, Washington, DC 20009, USA; 3Nutrition Section, UNICEF West and Central Africa Region, Dakar 29720, Senegal; 4Nutrition and Health, Action Contre la Faim, Dakar 29621, Senegal; 5Independent Researcher, Dakar 29621, Senegal

**Keywords:** severe stunting, stunting, HAZ scores, neonatal mortality, iron folic acid supplementation, malaria prophylaxis, sub-Saharan Africa, anemia during pregnancy

## Abstract

The majority of research on linear growth among children is confined to South Asia and focuses on iron and folic acid (IFA) supplementation during pregnancy, without considering malaria prophylaxis. Similarly, there is limited evidence on the association of antenatal IFA supplementation and malaria prophylaxis with neonatal mortality in sub-Saharan Africa (SSA). This study aims to address these gaps. A pooled analysis of demographic and health survey (DHS) data from 19 countries in SSA was conducted to study the association between IFA supplementation and malaria prophylaxis and linear growth and neonatal mortality. Multivariate logistic and linear regression models were used. Malaria prophylaxis was significantly associated with stunting, height-for-age Z scores (HAZ scores), and neonatal mortality, but IFA supplementation was not associated with these outcomes. When women’s height and body mass index (BMI) were introduced in the model, a significant association between combined malaria prophylaxis and IFA supplementation was found with HAZ scores only. For severe stunting, no significant association was found with either in the two models. In conclusion, this study underscores the importance of antenatal malaria prophylaxis as a potential intervention for nutrition outcomes (linear growth) and neonatal mortality, as well as the importance of coordinating efforts between malaria and the health and nutrition sectors to improve these outcomes in the countries of SSA.

## 1. Introduction

Africa accounts for two-fifths of stunting among children under five globally, ranking second to Asia in prevalence. While stunting in Africa has declined from 41.5% in 2000 to 30.7% in 2020, the absolute number of stunted children under five has increased in the same period, from 54.4 million to 61.4 million, respectively [1]. Neonatal mortality, on the other hand, is highest in Africa, at 27 deaths per 1000 live births [2]. Recent longitudinal studies in low- and middle-income countries (LMICs) have found that linear growth failure is highest at birth and in the first 3 months of life, and this early growth failure is associated with higher risk of persistent growth failure, low reversal rates, and child mortality [3,4]. Linear growth failure in early life originates mainly during intra uterine growth and is affected by maternal undernutrition. Low dietary intake, repeated infections, and rapid pregnancies directly contribute to maternal undernutrition and adverse pregnancy outcomes [5]. This highlights the importance of preventive interventions prior to and during pregnancy.

Multiple studies have shown a positive association of iron and folic acid (IFA) supplementation during pregnancy with fetal growth [6] and low birth weight [6,7,8,9,10,11,12,13]. However, very few have analyzed the association with a child’s linear growth beyond birth. Nisar et al. documented a reduction of 8% and 9% in adjusted risk of stunting and severe stunting respectively with antenatal IFA supplementation in seven South Asian countries [14]. In yet another study, Nisar et al. found a 14% lower risk of stunting among Nepalese children aged less than two years whose mothers reported antenatal IFA supplementation [15]. A study in a rural district of Indonesia, in which 35.7% of children under five were stunted, found anemia during pregnancy to be the most important factor associated with child stunting [16]. It is important to note that iron or hemoglobin status during pregnancy has a U-shaped association with birth outcomes. This indicates that the odds of stillbirth, preterm birth, and small size for gestational age are higher when maternal hemoglobin is very high or very low, and odds of poor birth outcomes are lower for middle values of hemoglobin between 9 and 13 g per deciliter [17]. Furthermore, the timing of pregnancy [18] and geographical context [19] have shown a moderating effect.

A systematic review and meta-analysis looked at adverse outcomes beyond birth and found an association between anemia during pregnancy and perinatal mortality (odds ratio of 2.9) in South Asia [8]. Another systematic review and meta-analysis in LMICs, conducted in 2016, concluded that 18% of perinatal mortality was attributable to anemia during pregnancy [11]. IFA supplementation during pregnancy has been associated with reduced neonatal mortality in Nepal [20], Pakistan [21], and Indonesia [22].

In summary, the majority of the research on IFA supplementation or anemia during pregnancy and child health outcomes has been conducted in South Asia. The situation is more complex in the African context due to the high prevalence of malaria. The region accounted for 94% of global malaria cases and deaths in 2019 [23]. Malaria during pregnancy has been found to be a risk factor for fetal growth retardation and neonatal mortality [24], while malaria prophylaxis during pregnancy has been associated with reduced risk of low birth weight [25].

In a 2010 study, Titaley et al. found that a combined regimen of IFA supplementation and malaria prophylaxis during pregnancy in nineteen countries of Sub-Saharan Africa (SSA) was associated with reduced neonatal mortality, but neither was effective alone [26]. To our knowledge, this is the only study that looked at the two regimens simultaneously in the African region. Furthermore, there are no studies that have looked at the association between this combined regimen and linear growth outcomes among children. This study aims to update and generate new evidence on the association of linear growth outcomes and neonatal mortality with antenatal exposure to both IFA supplementation and malaria prophylaxis in the SSA context.

## 2. Data and Methods

Nineteen countries in sub-Saharan Africa were selected for the pooled analysis. The inclusion criteria were (1) having a demographic and health survey (DHS) conducted in 2014 or later, and (2) surveys that included information on antenatal care (ANC) visits, IFA supplementation during pregnancy, sulfadoxine-pyrimethamine (SP)/Fansidar use during pregnancy, and anthropometric measurements for children under 5 years. The countries that met these criteria included Angola, Benin, Burundi, Cameroon, Chad, Gambia, Ghana, Guinea, Kenya, Liberia, Malawi, Mali, Mauritania, Nigeria, Senegal, Sierra Leone, Tanzania, Uganda, and Zambia.

For the analysis, the household member file and the birth recode file were first merged within each country and predictor variables were created. The country datasets were then pooled together.

The base population for the models related to linear growth included the youngest child under two years of age, while the base population for neonatal mortality included the youngest child between ages 1–23 months, to avoid censoring.

### 2.1. Outcome Variables

The following four outcomes were analyzed:Severe stunting: A binary variable that indicates children whose height-for-age z-score (HAZ score) were less than −3 standard deviations (SD) below the mean on the WHO Child Growth Standards.Stunting: A binary variable indicating that the child’s HAZ score was less than −2 SD below the mean on the WHO Child Growth Standards.HAZ score: A continuous variable that includes the z-score of children with a non-flagged height for age score.Neonatal mortality: A binary variable that indicates the death of a child within one month of birth.

### 2.2. Exposure Variables

These included the following main predictors and their combinations:IFA supplementation during pregnancy was divided into three categories based on the woman’s recall of number of days she consumed IFA during her last pregnancy: none, 1–90 days, and more than 90 days. Responses above 240 days were not included.Malaria prophylaxis during pregnancy indicated that the woman had taken at least two doses of SP/Fansidar when pregnant. This definition was based on the older WHO guidelines of 2014 to ensure consistency across countries. Some of the countries included in the study were surveyed in 2014 and included women who could have been pregnant 2–3 years earlier. Additionally, the implementation gap between WHO policy and national policy needed to be considered. Women who had no antenatal visit or were missing information on ANC and had responded in the affirmative for at least two doses, were assumed to not know about malaria prophylaxis and were not included.A composite variable on IFA supplementation and malaria prophylaxis was created and it categorized into combinations of the two variables in the following way: no IFA, no malaria prophylaxis; IFA 1 to 90 days, no malaria prophylaxis; IFA 90+ days, no malaria prophylaxis; no IFA, only malaria prophylaxis; IFA 1 to 90 days & malaria prophylaxis; and IFA 90+ days & malaria prophylaxis.In addition, a combined variable on IFA supplementation and timing of ANC visits was generated with the following categories: no iron or no ANC; 1st trimester ANC & IFA 1 to 90 days; 1st trimester ANC & IFA 90+ days; 2nd trimester ANC & IFA 1 to 90 days; 2nd trimester ANC & IFA 90+ days; and 3rd trimester ANC & IFA 1 to 90 days.

### 2.3. Other Independent Covariates

The following set of covariates were included in the multivariate models, along with country and malaria endemic zone. The latter is defined as the malaria transmission risk in an area and is based on number of plasmodium falciparum cases per thousand population at the provincial level within each country. A separate datafile was prepared using information from the Malaria Atlas Project (MAP) website [27] to merge with the provinces of study countries. The categorization of malaria case loads (MCL) was based on the cutoffs used in malaria heat maps by MAP, as follows: Grade 0/low endemic zone: 0 to <10 cases per thousand population; Grade 1: 10 to <175 cases per thousand population; Grade 2: 175 to <350 cases per thousand population; Grade 3: 350 to <700 cases per thousand population; and Grade 4: 700 or more cases per thousand population. Grades 0 and 1 were collapsed because of small cell sizes.

#### 2.3.1. Community Level and Socio-Economic Status

This included the following variables: improved sanitation (yes/no); improved water (yes/no); clean fuel for cooking (yes/no); household wealth (poorest, poorer, middle, richer, richest); place of residence (urban, rural); maternal education level (no education, primary, secondary or higher); and mother’s current marital status (married, unmarried).

#### 2.3.2. Maternal and Child Characteristics

This included the following variables: mother’s age at child birth (20 years or younger, 20–24 years, 25–49 years); birth order (1st, 2nd–3rd, 4th or higher); child’s gender (boy, girl); child’s age-group (<6 months, 6–8 months, 9–11 months, 12–17 months, 18–23 months); diarrhea in the 2 weeks before the survey (yes, no); number of children under five years (1, 2, 3 or more); early initiation of breastfeeding (child put to breast within one hour of birth); and proper nutrition (yes, no). The latter is a composite of two nutrition indicators (exclusive breastfeeding among infants under 6 months of age and minimum dietary diversity among children 6–23 months) to indicate adequate nutrition as per age. This variable was created to avoid loss of any child sub-population. Short height of mother (if mother’s height was less than 145 cm) and body mass index (BMI) of mother (underweight, healthy, overweight, obese) were also used.

#### 2.3.3. Number of ANC Visits (None, 1–3 Visits, 4 or More Visits)

##### Others

Time of recall was also considered, as it indicated the number of months that had elapsed since the month of interview until June 2022 (Range: 15–97; Mean: 60.16). Though it was used in the models initially, it was later discarded because of high multicollinearity (Variance Inflation Factor (VIF)-177).

For the neonatal mortality model, the following additional variables were considered: institutional delivery (yes, no); birth by C-section (yes, no); multiple pregnancies referring to the birth of twins, triplets, etc., as opposed to singleton pregnancy (yes, no); mother’s employment (yes, no); desire for pregnancy (wanted later/no more, wanted then); and low birth weight (LBW). LBW was categorized as smaller than average, average, and larger than average, as per the mother’s subjective perception. Information on birth weight was not available for 47.4% observations.A composite variable on parity and short birth interval was created to avoid loss of first-born children in multivariate analysis. The five categories were 1st birth; 2nd/3rd birth, <2 years interval; 4th or higher birth, <2 years interval; 2nd/3rd birth, ≥2 years interval; and 4th or higher birth, ≥2 years interval. Skilled birth assistance, created as per country specific definitions, was initially considered but was left out of the modelling because of high correlation with institutional delivery.

### 2.4. Methods

Descriptive statistics and multivariate analyses including linear regression (for HAZ scores) and logistic regression (stunting, severe stunting, neonatal mortality) were conducted using a multi-stage modelling process. In the first step, the variable set on community level and socio-economic status were included. This was followed by the variable set on maternal and child characteristics in the second step, and number of ANC visits in the third step. Backward elimination was carried out at each step and only variables significant at the *p*-value of 0.05 were kept. The last step included the exposure variables, which were introduced one at a time. Relevant interactions of exposure variables with timing of ANC and endemic zones could not be used because both predictors were not significant in any of the models. Country fixed effects were kept in all models. Poisson regression was also carried out for the binary outcomes but were deemed inferior to the logistic regression models based on the graphs using the calibration belt approach [28].

Separate modelling was conducted on a smaller pool of countries to include maternal short height and BMI. Three countries lacked this information; Angola, Senegal, and Zambia and were eliminated from the analysis. These models have been designated as Model B and have a smaller sample size because of missing information (across all countries).

The pooled data were weighted for multi-stage clustering and population size by adjusting the within-country weights by the mid-year population of women of reproductive age (15–49 years) in each country for the respective survey year (the earlier year if survey was conducted in two years). Survey setting was done with corrected stratum and primary sampling units for unique identification across countries. There was no multicollinearity, as the VIFs were less than 2.5. All results are presented as odds ratio with 95% confidence intervals.

The analyses were conducted using Stata 15.1. Table 1 shows the final sample sizes for the different multivariate models. Note that the prevalence of outcome and exposure variables is based on the study sample of most recent births under two years, while the distribution of other covariates is based on the final sample of Model A.

## 3. Results

Table 2 shows the country rankings on the prevalence and mean of the four outcome variables. As shown in the table, Ghana had the lowest prevalence for severe stunting and stunting, and the highest mean HAZ scores in the study countries, while the highest rates of stunting and the lowest mean HAZ were observed in Burundi. Burundi had the lowest prevalence of neonatal mortality, while Nigeria had the highest prevalence among the study countries.

Figure 1 shows the distribution of IFA consumption by number of days across study countries in the study sample. Burundi had the highest proportion of women reporting no IFA consumption at 57% and the lowest proportion of women reporting IFA consumption for more than 90 days at 1%. On the other hand, Zambia had one of the lowest proportions of women reporting no IFA consumption at 4% and the highest proportion of women reporting IFA consumption for more than 90 days at 61%. Countries in which 22–25% of women reported consuming IFA supplements for more than 90 days were also countries that reported a high proportion of women taking no IFA supplementation. These include Mali, Angola, Mauritania, and Nigeria. In Uganda, Tanzania, Malawi, Guinea, Kenya, and Sierra Leone, over 60% of women reported consuming IFA supplements between 1–90 days.

Figure 2 shows the distribution of at least two doses of malaria prophylaxis by country in the study sample. The coverage of malaria prophylaxis among pregnant women was lowest in Kenya, Chad, Burundi, and Mauritania, with less than one-quarter of the women in the study population consuming malaria prophylaxis. Coverage was higher in Liberia, Sierra Leone, Gambia, and Zambia, with more than 70% of women consuming malaria prophylaxis. Note that the middle order countries, such as Uganda, Cameroon, and Mali, had almost 50% coverage.

Table 3 shows the distribution of outcomes across exposure variables. The prevalence of severe stunting, stunting, and neonatal mortality became significantly lower with increasing supplementation of IFA as compared to no supplementation, and with supplementation of at least two doses of malaria prophylaxis during pregnancy. When IFA supplementation as well as timing of first ANC visit were considered, the lowest prevalence of severe stunting and stunting was among mothers who had more than 90 days of supplementation starting in the first trimester and the highest was among those who had neither IFA supplementation nor ANC visits. When looking at IFA supplement consumption and malaria prophylaxis simultaneously, the lowest prevalence of severe stunting and stunting was among mothers who had consumed IFA supplements for more than 90 days along with malaria prophylaxis and the highest among those who had taken neither. Mean HAZ scores had the same distribution as severe stunting and stunting but in an opposite direction.

In case of neonatal mortality, the highest prevalence of mortality was among mothers who had neither consumed IFA supplements nor made ANC visits and the lowest was among those who had taken between 1 to 90 days of supplements starting in the third trimester. Prevalence of neonatal mortality was lowest among mothers who had between 1 to 90 days of IFA supplementation along with malaria prophylaxis and was highest among those who had neither.

Table 4 shows the prevalence and mean of outcome variables by select characteristics in the model sample. As seen, the prevalence of severe stunting and stunting were highest among the poorest, adolescent mothers, mothers with no education, mothers who had no ANC visits, boys, children in the age-group 18–23 months, children who had diarrhea in the past two weeks, children who had no proper nutrition, those living in a Grade 4 endemic zone, and those belonging to higher birth orders as compared to their counterparts. The prevalence of severe stunting was almost three times higher and that of stunting was almost double in mothers with short height as compared to those of average height. The prevalence of severe stunting and stunting was high among mothers with undernutrition. The opposite was true for mean HAZ scores.

Prevalence of neonatal mortality was highest among infants from the middle wealth quintile, among adolescent mothers, among mothers who had no ANC visits, among boys, and when a woman had three or more children under 5.

### 3.1. Multivariate Findings

#### 3.1.1. Severe Stunting

Table 5 shows the odds ratios and the 95% confidence intervals (CI) from the two final multivariate models; Model A that does not include mother’s anthropometric indicators and Model B that does, for severe stunting. After controlling for multiple factors, the significant negative association at the bivariate level between IFA supplementation during pregnancy and severe stunting among children under two became non-significant. No significant association was found between malaria prophylaxis during pregnancy and severe stunting among children under two.

#### 3.1.2. Stunting

Table 6 shows findings from the two final multivariate models; Model A that does not include mother’s anthropometric indicators and Model B that does, for stunting. After controlling for multiple factors, the significant negative association at the bivariate level between supplementation of iron during pregnancy and stunting among children under two was lost. However, a significant association was found between malaria prophylaxis during pregnancy and stunting among children under two, where the odds of stunting were 10% lower among women who had malaria prophylaxis during pregnancy.

#### 3.1.3. HAZ Scores

Table 7 shows the coefficients and 95% CI from the two final multivariate models for HAZ scores. Model A does not include mother’s anthropometric indicators and Model B does. After controlling for other variables, a significant positive association was found between malaria prophylaxis during pregnancy and mean HAZ scores among children under two, where the mean HAZ scores were higher by 0.058 among women who had malaria prophylaxis during pregnancy. When maternal height and BMI were additionally controlled for (Model B), a significant positive association was found with combined IFA supplementation of more than 90 days and malaria prophylaxis. Children born to women who reported receiving both IFA (>90 days) and malaria prophylaxis during pregnancy had a significantly higher HAZ (0.104) as compared to children born to women who had received neither IFA supplementation nor malaria prophylaxis.

#### 3.1.4. Neonatal Mortality

Table 8 shows the two final multivariate models for neonatal mortality. Model A, which does not include mother’s anthropometric indicators, and Model B, which does. After controlling for multiple factors, the significant negative association at the bivariate level between supplementation of iron during pregnancy and neonatal mortality among children 1–23 months old was lost. However, a significant negative association was found between malaria prophylaxis during pregnancy and neonatal mortality in this population. Neonatal mortality had 20% lower odds among mothers who had received malaria prophylaxis during pregnancy. When maternal height and BMI were additionally controlled for (Model B), the odds for neonatal mortality were lower by almost 30% (95% CI: 0.565, 0.914) among mothers who had received malaria prophylaxis during pregnancy.

## 4. Discussion

Overall, this study found significant association of malaria prophylaxis with stunting, HAZ scores, and neonatal mortality, but there was no association of these outcomes with IFA supplementation. When women’s anthropometric indicators such as height and BMI were introduced, a significant positive association of combined malaria prophylaxis and IFA supplementation was found with HAZ scores, but only malaria prophylaxis remained significantly and negatively associated with stunting and neonatal mortality. For severe stunting, no significant association was found with either malaria prophylaxis or IFA supplementation in either of the models.

The findings from this study diverge from the results of similar studies in South Asia, where IFA supplementation and the timing of its initiation were found to be significantly associated with linear growth [13,14] and neonatal mortality [7,19,20,21]. This contrast that also highlights the importance of context may be explained by the high prevalence of malaria in SSA. The latter translates to high prevalence of the infection during pregnancy; the complex interrelationship between iron status and malaria; and interactions of IFA with malaria prophylaxis.

Malaria infection increases the risk of anemia by destroying red blood cells and reducing their production [29]. It also reduces iron absorption, which reduces the efficacy of iron supplementation [30]. On the other hand, IFA supplementation increases malaria risk and has been associated with increased placental parasitemia, while the opposite is true for iron deficiency [30]. Furthermore, the amount of folic acid in the IFA supplement impacts the effectiveness of malaria prophylaxis regimen differently. For example, if given concurrently, 5 mg of folic acid renders malaria prophylaxis ineffective, while 0.4 mg of folic acid shows no interference [31]. This nexus is further affected by host immunity, which tends to be high in endemic areas and protects against malaria [30].

Given this complex interrelationship between malaria, anemia, and IFA supplementation, any positive association of IFA supplementation with the study outcomes in SSA may have been overridden by malaria infection, thereby allowing only the association with malaria prophylaxis to stand out. These findings are corroborated by Titaley and Dibley [26], who noted that the protective effect of IFA supplementation on neonatal mortality was lower in the SSA region as compared to countries outside the region. The authors attributed this to the presence of inflammation (malaria and HIV) among pregnant women.

However, Titaley and Dibley [26] also found a significant association between neonatal mortality and a combination of IFA supplementation and malaria prophylaxis in SSA, but not when either of the two exposure variables were considered alone. Our study showed similar findings, but only for HAZ scores, and only when the models were controlled for maternal BMI. This difference can be explained by the fact that a continuous outcome, as opposed to a binary outcome, increases the statistical power and allows detection of smaller effect sizes (HAZ scores versus severe stunting and stunting), whereas controlling for maternal BMI further improved the validity of results. This result also indicates that the combined regimen of antenatal IFA supplementation and malaria prophylaxis seems to be more beneficial in preventing adverse birth outcomes among women with poor nutritional status.

Therefore, policymakers in SSA should consider actions that would strengthen the integration of IFA interventions with malaria prophylaxis during pregnancy and aim to improve their coverage [32]. Burundi stands out among the study countries with the highest prevalence of poor linear growth among children under 5; the lowest prevalence of IFA consumption of more than 90 days at 1%; and only 21% prevalence of consumption of two doses of malaria prophylaxis. An improvement in coverage of both interventions may help reduce linear growth failure among children under five. In comparison, for Nigeria–with the highest prevalence of neonatal mortality among study countries–the focus needs to be on improvement in the coverage of malaria prophylaxis. In addition, there is some evidence indicating that some countries continue to use 5 mg of folic acid [31], and this should be rectified.

The study is not without limitations and the findings should be reviewed accordingly. (1) There is possibility of recall bias in the iron consumption information. We restricted the sample to the most recent child under two years to help reduce this bias. (2) Iron consumption information showed heaping at monthly intervals, and hence, there was a possibility of misclassification. This was addressed to a big extent by categorization into two categories. (3) The pooled dataset did not include data from all countries of sub-Saharan Africa and those that were included had been surveyed in different years. So, the pooled data was not from a well-defined population and cannot be considered as representative of sub-Saharan Africa. (4) The post-estimation statistics for some models were not ideal, but this could also be because of the large sample size. (5) The possibility of endogeneity cannot be ruled out. (6) It was assumed that two doses of malaria prophylaxis during pregnancy could not be consumed by a pregnant woman unless she had an antenatal visit, which may not be true, and could have caused some misclassification. However, such observations were very few and amounted to less than two percent of the total observations on malaria prophylaxis.

## 5. Conclusions

This study provides new evidence that children experience greater linear growth when their mothers consume both IFA supplementation and malaria prophylaxis during pregnancy in the SSA context. In particular, the study highlights the importance of malaria prophylaxis during pregnancy as a potential influence for decreasing stunting among children and neonatal mortality in the countries of SSA. This study also underscores the importance of coordinated efforts between malaria and the health and nutrition sectors to improve these outcomes. Policymakers in SSA should aim to improve the coverage of malaria prophylaxis and IFA supplementation during pregnancy and consider how to strengthen the integration between the two. Since this study has shown that context matters, similar analysis can be applied to other countries or regions for better understanding of the situation in the local context.

## Figures and Tables

**Figure 1 nutrients-14-04496-f001:**
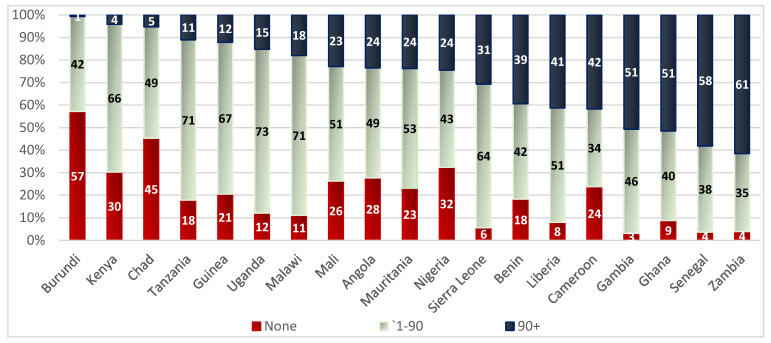
Distribution of IFA consumption (number of days) by country among mothers of youngest child under two years of age.

**Figure 2 nutrients-14-04496-f002:**
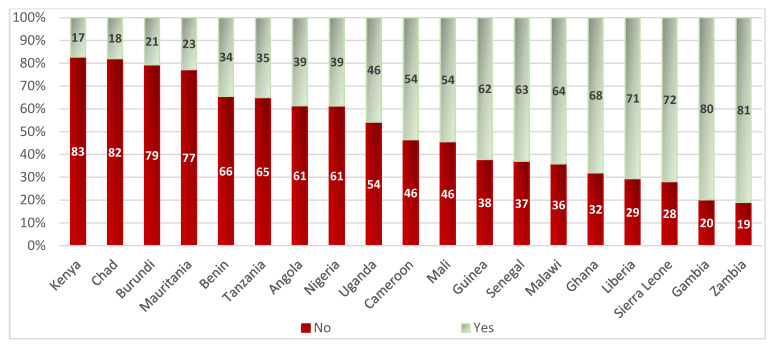
Distribution of malaria prophylaxis (at least two doses) by country among mothers of youngest child under two years of age.

**Table 1 nutrients-14-04496-t001:** Sample size of the final models.

	Study Sample	Model A ^#^	Model B ^##^
Severe stunting	56,388	55,508	32,909
Stunting	56,388	54,779	36,462
HAZ scores	56,388	54,779	33,892
Neonatal mortality	90,503	82,203	37,482

^#^ Model A: includes all 19 countries. **^##^** Model B: includes 16 countries that had information on mother’s anthropometry.

**Table 2 nutrients-14-04496-t002:** Country ranking by prevalence and mean for the four outcomes among youngest child under two years of age.

Severe Stunting		Stunting		HAZ Scores		Neonatal Mortality	
Country	%	Country	%	Country	Mean	Country	%
Ghana	3.2	Ghana	11.8	Ghana	−0.6	Burundi	1.61
Gambia	3.6	Gambia	16.5	Cameroon	−0.8	Kenya	1.70
Senegal	5.1	Senegal	17.3	Mali	−0.8	Ghana	1.85
Mauritania	5.7	Mauritania	20.3	Chad	−0.9	Zambia	1.89
Liberia	5.8	Kenya	21.1	Guinea	−0.9	Senegal	1.92
Kenya	6.6	Mali	21.6	Kenya	−0.9	Mauritania	2.03
Benin	7.1	Liberia	23.5	Mauritania	−0.9	Tanzania	2.13
Malawi	7.5	Sierra Leone	23.9	Senegal	−0.9	Cameroon	2.31
Uganda	7.5	Uganda	24.2	Uganda	−0.9	Angola	2.32
Mali	7.8	Benin	24.4	Gambia	−1.0	Gambia	2.33
Sierra Leone	8.1	Chad	25.1	Sierra Leone	−1.1	Mali	2.35
Tanzania	9.0	Cameroon	26.6	Tanzania	−1.1	Uganda	2.38
Nigeria	11.1	Tanzania	26.7	Benin	−1.2	Malawi	2.39
Chad	11.2	Guinea	27.3	Liberia	−1.2	Benin	2.46
Zambia	11.7	Nigeria	29.4	Malawi	−1.2	Liberia	2.59
Cameroon	12.3	Malawi	30.0	Nigeria	−1.2	Sierra Leone	2.59
Angola	12.7	Zambia	30.9	Angola	−1.3	Chad	2.67
Guinea	12.8	Angola	31.3	Zambia	−1.3	Guinea	3.00
Burundi	16.0	Burundi	45.3	Burundi	−1.8	Nigeria	3.18

**Table 3 nutrients-14-04496-t003:** Distribution of the exposure variables by outcomes.

Exposure Variables	Severe Stunting	Stunting	HAZ (Mean) ^@^	NM ^@^
**Number of days IFA consumed**	***	***		*
None	12.69	31.81	−1.24	2.87
1–90	9.08	25.89	−1.07	2.26
90+	7.36	21.99	−1.00	2.26
**Two or more doses of SP/Fansidar**	***	***		*
No	9.78	27.60	−1.12	2.62
Yes	8.06	22.79	−0.99	2.07
**Days IFA consumed + timing of ANC visit**	***	***		*
No iron/No ANC	12.62	31.52	−1.24	2.89
1st trimester_1 to 90	7.48	23.45	−1.00	2.41
1st trimester_90+	6.21	20.11	−0.94	2.34
2nd trimester_1 to 90	9.23	26.59	−1.09	2.24
2nd trimester_90+	8.45	23.75	−1.07	2.14
3rd trimester_1 to 90	11.52	27.82	−1.15	1.97
**IFA + Malaria prophylaxis (IPTP2)**	***	***		*
No IFA, No IPTP2	13.03	32.50	−1.26	2.94
IFA 1–90, No IPTP2	9.39	27.34	−1.10	2.48
IFA 90+, No IPTP2	7.61	24.32	−1.07	2.41
No IFA, Only IPTP2	10.75	27.82	−1.11	2.42
IFA 1–90 & IPTP2	8.63	24.06	−1.02	2.01
IFA 90+ & IPTP2	7.24	20.92	−0.97	2.18

^@^ HAZ: Height–for–age Z–scores; NM: Neonatal mortality. *p*–value: * *p* < 0.05, *** *p* < 0.000.

**Table 4 nutrients-14-04496-t004:** Prevalence and mean of outcome variables by select characteristics.

	Severe Stunting	Stunting	HAZ Scores	Neonatal Mortality
	Row%	Row%	Mean	Row%
**Household wealth**	***	***		*
Poorest	12.11	31.51	−1.2	2.07
Poorer	11.06	29.87	−1.2	2.65
Middle	8.39	25.76	−1.1	2.85
Richer	8.23	22.63	−1	2.27
Richest	4.31	15.31	−0.7	1.94
**Mother’s age**	***	***		***
<20	10.88	29.82	−1.2	3.17
20–24	8.79	26.00	−1.1	1.77
25–49	8.72	24.20	−1	2.43
**Mother’s education**	***	***		
No education	12.43	30.82	−1.1	2.53
Primary	9.26	26.77	−1.1	2.42
Secondary or higher	5.32	18.59	−0.8	2.16
**Number of ANC visits**	***	***		***
None	15.11	34.88	−1.2	3.38
1–3 visits	10.00	27.60	−1.1	2.38
4 or more visits	7.46	22.81	−1	2.16
**Child’s gender**	***	***		*
Boy	10.88	29.10	−1.2	2.64
Girl	7.17	21.80	−0.9	2.09
**Child’s age–group**	***	***		
<6 months	5.16	14.59	−0.5	
6–8 months	5.34	16.36	−0.7	
9–11 months	6.71	21.11	−1	
12–17 months	10.80	30.80	−1.3	
18–23 months	15.12	40.09	−1.7	
**Children under 5**	**	***		***
1	8.28	24.48	−1.1	2.28
2	9.78	26.66	−1.1	2.08
3 or more	9.01	24.51	−1	4.29

*p*-value: * *p* < 0.05, ** *p* < 0.001, *** *p* < 0.000.

**Table 5 nutrients-14-04496-t005:** Multivariate logistic regression model on severe stunting.

	Model A		Model B	
	OR	95% C.I.	OR	95% C.I.
**Country**				
Angola	2.216 ***	[1.757, 2.794]	na	
Benin ^R^	1	[1, 1]	1	[1, 1]
Burundi	2.789 ***	[2.333, 3.333]	2.564 ***	[2.071, 3.176]
Cameroon	2.184 ***	[1.757, 2.717]	2.338 ***	[1.820, 3.003]
Chad	1.482 ***	[1.238, 1.774]	1.545 ***	[1.248, 1.912]
Gambia	0.594 **	[0.418, 0.845]	0.632 *	[0.435, 0.918]
Ghana	0.568 *	[0.350, 0.922]	0.571 *	[0.342, 0.951]
Guinea	1.829 ***	[1.447, 2.312]	1.879 ***	[1.446, 2.443]
Kenya	1.097	[0.907, 1.327]	1.227	[0.944, 1.596]
Liberia	1.036	[0.712, 1.509]	1.044	[0.702, 1.553]
Malawi	1.265	[0.997, 1.607]	1.234	[0.942, 1.615]
Mali	1.020	[0.845, 1.232]	1.143	[0.878, 1.487]
Mauritania	0.838	[0.675, 1.041]	0.942	[0.710, 1.250]
Nigeria	1.896 ***	[1.577, 2.280]	1.984 ***	[1.591, 2.475]
Senegal	0.699 *	[0.523, 0.935]	na	
Sierra Leone	1.283	[0.971, 1.695]	1.26	[0.929, 1.711]
Tanzania	1.413 ***	[1.169, 1.709]	1.442 **	[1.151, 1.805]
Uganda	1.332 *	[1.042, 1.702]	1.353 *	[1.025, 1.785]
Zambia	2.307 ***	[1.911, 2.786]	na	
**Household wealth**				
Poorest ^R^	1	[1, 1]	1	[1, 1]
Poorer	0.956	[0.848, 1.078]	1.005	[0.865, 1.167]
Middle	0.796 ***	[0.695, 0.911]	0.810 *	[0.683, 0.961]
Richer	0.886	[0.744, 1.056]	0.905	[0.730, 1.120]
Richest	0.530 ***	[0.429, 0.655]	0.582 ***	[0.449, 0.755]
**Mother’s education**				
No education ^R^	1	[1, 1]	1	[1, 1]
Primary	0.752 ***	[0.665, 0.849]	0.719 ***	[0.619, 0.837]
Secondary/Higher	0.488 ***	[0.419, 0.568]	0.453 ***	[0.375, 0.549]
**Child’s age–group**				
<6 months ^R^	1	[1, 1]	1	[1, 1]
6–8 months	1.06	[0.870, 1.292]	1.089	[0.853, 1.390]
9–11 months	1.427 ***	[1.180, 1.726]	1.511 ***	[1.191, 1.918]
12–17 months	2.322 ***	[1.996, 2.703]	2.448 ***	[2.029, 2.953]
18–23 months	3.661 ***	[3.166, 4.233]	3.894 ***	[3.260, 4.652]
**Mother’s age–group**				
<20 ^R^	1	[1, 1]	1	[1, 1]
20–24	0.798 **	[0.688, 0.926]	0.844	[0.699, 1.017]
25–49	0.757 ***	[0.658, 0.870]	0.758 **	[0.638, 0.902]
**Gender**				
Boy ^R^	1	[1, 1]	1	[1, 1]
Girl	0.626 ***	[0.569, 0.688]	0.649 ***	[0.577, 0.730]
**Children under 5**				
1 ^R^	1	[1, 1]	1	[1, 1]
2	1.120 *	[1.003, 1.250]	1.115	[0.972, 1.279]
3 or more	1.237 *	[1.039, 1.473]	1.257 *	[1.019, 1.550]
**Number of ANC visits**				
None ^R^	1	[1, 1]	1	[1, 1]
1–3 visits	0.889	[0.761, 1.039]	0.911	[0.756, 1.098]
4 or more visits	0.725 ***	[0.619, 0.849]	0.739 **	[0.610, 0.896]
**Short height**				
No ^R^			1	[1, 1]
Yes			3.676 ***	[2.665, 5.071]
**BMI**				
Underweight ^R^			1	[1, 1]
Healthy			0.868	[0.738, 1.022]
Overweight			0.762 *	[0.588, 0.988]
Obese			0.823	[0.598, 1.134]
**Early initiation of BF**				
No ^R^	1	[1, 1]		
Yes	0.899 *	[0.812, 0.996]		

* *p* < 0.05, ** *p* < 0.01, *** *p* < 0.001. CI: Confidence Interval; ^R^: Reference category; na: Information not available. Model A: Includes all 19 countries. Model B: Includes 16 countries that had information on mother’s anthropometry.

**Table 6 nutrients-14-04496-t006:** Multivariate logistic regression model on stunting.

	Model A		Model B	
	OR	95% C.I.	OR	95% C.I.
**Malaria prophylaxis**				
No ^R^	1	[1, 1]	1	[1, 1]
Yes	0.901 **	[0.837, 0.970]	0.904 *	[0.826, 0.989]
**Country**				
Angola	1.523 ***	[1.293, 1.794]	na	
Benin ^R^	1	[1, 1]	1	[1, 1]
Burundi	2.471 ***	[2.124, 2.875]	2.253 ***	[1.877, 2.706]
Cameroon	1.176	[0.978, 1.414]	1.282 *	[1.046, 1.572]
Chad	0.791 **	[0.672, 0.931]	0.722 **	[0.593, 0.878]
Gambia	0.591 ***	[0.466, 0.750]	0.565 ***	[0.433, 0.737]
Ghana	0.481 ***	[0.380, 0.609]	0.506 ***	[0.395, 0.649]
Guinea	1.119	[0.929, 1.347]	1.129	[0.920, 1.387]
Kenya	0.749 ***	[0.637, 0.879]	0.757 *	[0.611, 0.938]
Liberia	1.094	[0.852, 1.405]	1.082	[0.825, 1.419]
Malawi	1.359 **	[1.111, 1.662]	1.471 ***	[1.177, 1.839]
Mali	0.748 ***	[0.642, 0.872]	0.810 *	[0.665, 0.987]
Mauritani	0.666 ***	[0.556, 0.797]	0.653 ***	[0.511, 0.833]
Nigeria	1.380 ***	[1.144, 1.666]	1.497 ***	[1.211, 1.851]
Senegal	0.552 ***	[0.437, 0.697]	na	
Sierra Leone	1.109	[0.927, 1.326]	1.136	[0.931, 1.386]
Tanzania	1.095	[0.957, 1.252]	1.089	[0.927, 1.279]
Uganda	1.013	[0.816, 1.258]	1.126	[0.887, 1.429]
Zambia	1.605 ***	[1.388, 1.855]	na	
**Endemic zone**				
Grade 0/1 ^R^	1	[1, 1]	1	[1, 1]
Grade 2	0.841 **	[0.747, 0.947]	0.801 **	[0.687, 0.934]
Grade 3	0.837 **	[0.736, 0.953]	0.777 **	[0.662, 0.913]
Grade 4	0.901	[0.756, 1.073]	0.779 *	[0.638, 0.951]
**Household wealth**				
Poorest ^R^	1	[1, 1]	1	[1, 1]
Poorer	0.957	[0.879, 1.041]	0.98	[0.879, 1.094]
Middle	0.842 ***	[0.767, 0.925]	0.904	[0.803, 1.017]
Richer	0.762 ***	[0.681, 0.852]	0.809 **	[0.704, 0.929]
Richest	0.530 ***	[0.462, 0.608]	0.587 ***	[0.497, 0.693]
**Mother’s education**				
No education ^R^	1	[1, 1]	1	[1, 1]
Primary	0.827 ***	[0.762, 0.899]	0.774 ***	[0.698, 0.857]
Secondary/Higher	0.624 ***	[0.564, 0.690]	0.581 ***	[0.514, 0.658]
**Child’s age–group**				
<6 months ^R^	1	[1, 1]	1	[1, 1]
6–8 months	1.104	[0.975, 1.249]	1.112	[0.954, 1.297]
9–11 months	1.605 ***	[1.422, 1.811]	1.674 ***	[1.437, 1.950]
12–17 months	2.703 ***	[2.451, 2.982]	2.810 ***	[2.488, 3.174]
18–23 months	4.374 ***	[3.960, 4.832]	4.574 ***	[4.036, 5.184]
**Mother’s age–group**				
<20 ^R^	1	[1, 1]	1	[1, 1]
20–24	0.877 *	[0.788, 0.975]	0.861 *	[0.753, 0.986]
25–49	0.782 ***	[0.692, 0.885]	0.796 **	[0.682, 0.929]
**Proper nutrition**				
No ^R^	1	[1, 1]	1	[1, 1]
Yes	0.910 *	[0.846, 0.979]	0.904 *	[0.827, 0.988]
**Diarrhea in past 2 weeks**				
No ^R^	1	[1, 1]	1	[1, 1]
Yes	1.150 ***	[1.072, 1.233]	1.148 **	[1.051, 1.254]
**Gender**				
Boy ^R^	1	[1, 1]	1	[1, 1]
Girl	0.658 ***	[0.620, 0.698]	0.682 ***	[0.634, 0.734]
**Children under 5**				
1 ^R^	1	[1, 1]	1	[1, 1]
2	1.207 ***	[1.107, 1.315]	1.232 ***	[1.107, 1.372]
3 or more	1.370 ***	[1.206, 1.558]	1.438 ***	[1.229, 1.683]
**Birth order**				
1st ^R^	1	[1, 1]	1	[1, 1]
2nd–3rd	0.778 ***	[0.694, 0.872]	0.757 ***	[0.657, 0.871]
4th or higher	0.847 *	[0.736, 0.975]	0.814 *	[0.685, 0.968]
**Number of ANC visits**				
None ^R^	1	[1, 1]	1	[1, 1]
1–3 visits	0.956	[0.853, 1.071]	0.935	[0.813, 1.076]
4 or more visits	0.825 ***	[0.736, 0.924]	0.814 **	[0.708, 0.937]
**Short height**				
No ^R^			1	[1, 1]
Yes			2.938 ***	[2.202, 3.920]
**BMI**				
Underweight ^R^			1	[1, 1]
Healthy			0.860 *	[0.759, 0.975]
Overweight			0.727 ***	[0.612, 0.864]
Obese			0.733 **	[0.588, 0.913]

* *p* < 0.05, ** *p* < 0.01, *** *p* < 0.001. CI: Confidence Interval; ^R^: Reference category; na: Information not available. Model A: Includes all 19 countries. Model B: Includes 16 countries that had information on mother’s anthropometry.

**Table 7 nutrients-14-04496-t007:** Multivariate linear regression model on HAZ scores.

	Model A		Model B	
	Coeff.	95% C.I.	Coeff.	95% C.I.
**Malaria prophylaxis**				
No ^R^	0	[0, 0]		
Yes	0.058 **	[0.015, 0.102]		
**IFA + Malaria prophylaxis (IPTP2)**				
No IFA, No IPTP2 ^R^			0	[0, 0]
IFA 1–90, No IPTP2			0.034	[−0.042, 0.110]
IFA 90+, No IPTP2			−0.002	[−0.111, 0.107]
No IFA, Only IPTP2			0.024	[−0.139, 0.187]
IFA 1–90 & IPTP2			0.056	[−0.028, 0.141]
IFA 90+ & IPTP2			0.106 *	[0.006, 0.206]
**Country**				
Angola	−0.219 ***	[−0.324, −0.114]	na	
Benin ^R^	0	[0, 0]	0	[0, 0]
Burund	−0.568 ***	[−0.652, −0.485]	−0.502 ***	[−0.612, −0.393]
Cameroon	0.209 ***	[0.087, 0.331]	0.135	[−0.004, 0.273]
Chad	0.538 ***	[0.440, 0.636]	0.595 ***	[0.475, 0.715]
Gambia	0.151 **	[0.040, 0.261]	0.185 **	[0.058, 0.313]
Ghana	0.450 ***	[0.344, 0.557]	0.413 ***	[0.292, 0.534]
Guinea	0.350 ***	[0.208, 0.492]	0.366 ***	[0.211, 0.521]
Kenya	0.387 ***	[0.300, 0.474]	0.346 ***	[0.226, 0.467]
Liberia	−0.095	[−0.233, 0.042]	−0.0521	[−0.203, 0.099]
Malawi	−0.040	[−0.160, 0.080]	−0.0988	[−0.237, 0.039]
Mali	0.416 ***	[0.323, 0.510]	0.370 ***	[0.239, 0.500]
Mauritania	0.286 ***	[0.183, 0.388]	0.307 ***	[0.168, 0.446]
Nigeria	−0.152 **	[−0.261, −0.043]	−0.185 **	[−0.316, −0.054]
Senegal	0.353 ***	[0.243, 0.463]	na	
Sierra Leone	0.040	[−0.065, 0.144]	0.032	[−0.088, 0.152]
Tanzania	0.107 **	[0.029, 0.185]	0.116 *	[0.019, 0.213]
Uganda	0.237 ***	[0.109, 0.365]	0.158 *	[0.010, 0.306]
Zambia	−0.162 ***	[−0.249, −0.075]	na	
**Endemic zone**				
Grade 0/1 ^R^	0	[0, 0]	0	[0, 0]
Grade 2	0.098 **	[0.032, 0.164]	0.134 **	[0.047, 0.221]
Grade 3	0.117 **	[0.043, 0.190]	0.152 ***	[0.063, 0.241]
Grade 4	0.094	[−0.008, 0.195]	0.180 **	[0.057, 0.303]
**Household wealth**				
Poorest ^R^	0	[0, 0]	0	[0, 0]
Poorer	0.014	[−0.043, 0.071]	−0.021	[−0.096, 0.053]
Middle	0.095 **	[0.037, 0.152]	0.026	[−0.047, 0.098]
Richer	0.152 ***	[0.086, 0.218]	0.102 *	[0.020, 0.185]
Richest	0.356 ***	[0.275, 0.437]	0.295 ***	[0.203, 0.388]
**Mother’s education**				
No education ^R^	0	[0, 0]	0	[0, 0]
Primary	0.063 *	[0.013, 0.112]	0.110 ***	[0.0471, 0.172]
Secondary/Higher	0.210 ***	[0.152, 0.267]	0.282 ***	[0.209, 0.356]
**Currently married**				
No ^R^	0	[0, 0]	0	[0, 0]
Yes	0.064 *	[0.008, 0.119]	0.088 *	[0.017, 0.160]
**Child’s age–group**				
<6 months ^R^	0	[0, 0]	0	[0, 0]
6–8 months	−0.130 ***	[−0.192, −0.068]	−0.176 ***	[−0.253, −0.100]
9–11 months	−0.422 ***	[−0.484, −0.360]	−0.486 ***	[−0.566, −0.405]
12–17 months	−0.773 ***	[−0.825, −0.721]	−0.847 ***	[−0.913, −0.781]
18–23 months	−1.143 ***	[−1.195, −1.090]	−1.228 ***	[−1.295, −1.161]
**Mother’s age–group**				
<20 ^R^	0	[0, 0]	0	[0, 0]
20–24	0.114 ***	[0.053, 0.175]	0.109 **	[0.033, 0.186]
25–49	0.198 ***	[0.127, 0.269]	0.189 ***	[0.099, 0.278]
**Gender**				
Boy ^R^	0	[0, 0]	0	[0, 0]
Girl	0.239 ***	[0.203, 0.275]	0.222 ***	[0.177, 0.267]
**Children under 5**				
1 ^R^	0	[0, 0]	0	[0, 0]
2	−0.093	[−0.142, −0.043]	−0.095 **	[−0.157, −0.034]
3 or more	−0.162 ***	[−0.235, −0.089]	−0.175 ***	[−0.265, −0.084]
**Birth order**				
1st ^R^	0	[0, 0]	0	[0, 0]
2nd	0.115 ***	[0.049, 0.180]	0.131 **	[0.049, 0.213]
3rd or higher	0.03	[−0.050, 0.109]	0.034	[−0.067, 0.135]
**Number of ANC visits**				
None ^R^	0	[0, 0]	0	[0, 0]
1–3 visits	0.021	[−0.058, 0.100]	0.006	[−0.106, 0.117]
4 or more visits	0.136 ***	[0.057, 0.215]	0.118 *	[0.003, 0.233]
**Clean fuel for cooking**				
No ^R^	0	[0, 0]		
Yes	0.148 ***	[0.065, 0.232]		
**Area of residence**				
Urban ^R^	0	[0, 0]		
Rural	−0.055	[−0.105, −0.005]		
**Diarrhea in past 2 weeks**				
No ^R^	0	[0, 0]		
Yes	−0.088	[−0.130, −0.046]		
**Short height**				
No ^R^			0	[0, 0]
Yes			−0.807 ***	[−1.019, −0.595]
**BMI**				
Underweight ^R^			0	[0, 0]
Healthy			0.118 **	[0.0374, 0.198]
Overweight			0.281 ***	[0.181, 0.381]
Obese			0.301 ***	[0.182, 0.420]

* *p* < 0.05, ** *p* < 0.01, *** *p* < 0.001. CI: Confidence Interval; ^R^: Reference category; na: Information not available. Model A: Includes all 19 countries. Model B: Includes 16 countries that had information on mother’s anthropometry.

**Table 8 nutrients-14-04496-t008:** Multivariate logistic regression model on neonatal mortality.

Neonatal Mortality	Model A		Model B	
	OR	95% C.I.	OR	95% C.I.
**Malaria prophylaxis**				
No ^R^	1	[1, 1]	1	[1, 1]
Yes	0.811 *	[0.690, 0.954]	0.719 **	[0.565, 0.914]
**Country**				
Benin ^R^	1	[1, 1]	1	[1, 1]
Angola	1.085	[0.765, 1.538]	na	
Burundi	1.220	[0.859, 1.734]	1.902 **	[1.192, 3.035]
Cameroon	0.927	[0.653, 1.316]	0.905	[0.547, 1.500]
Chad	0.858	[0.620, 1.185]	0.862	[0.569, 1.308]
Gambia	0.979	[0.673, 1.425]	1.118	[0.667, 1.874]
Ghana	0.981	[0.638, 1.509]	1.070	[0.592, 1.932]
Guinea	1.368	[0.971, 1.925]	1.387	[0.844, 2.281]
Kenya	0.847	[0.557, 1.288]	0.990	[0.607, 1.616]
Liberia	1.722 **	[1.158, 2.562]	1.877	[0.974, 3.620]
Malawi	1.687 **	[1.223, 2.327]	2.071 **	[1.226, 3.497]
Mali	1.220	[0.869, 1.713]	1.769 *	[1.103, 2.839]
Mauritania	0.802	[0.558, 1.152]	0.845	[0.496, 1.441]
Nigeria	1.320 *	[1.005, 1.733]	1.525 *	[1.044, 2.229]
Senegal	0.543 **	[0.357, 0.826]	na	
Sierra Leone	1.544 *	[1.066, 2.237]	2.285 **	[1.373, 3.801]
Tanzania	1.089	[0.754, 1.573]	1.326	[0.864, 2.035]
Uganda	1.345	[0.991, 1.825]	1.603 *	[1.007, 2.553]
Zambia	1.429	[0.982, 2.079]	na	
**Household wealth**				
Poorest ^R^	1	[1, 1]	1	[1, 1]
Poorer	1.326 **	[1.076, 1.634]	1.163	[0.833, 1.626]
Middle	1.496 ***	[1.207, 1.855]	1.498 *	[1.085, 2.069]
Richer	1.302 *	[1.033, 1.643]	1.159	[0.826, 1.628]
Richest	1.107	[0.842, 1.455]	0.918	[0.589, 1.430]
**Mother’s age–group**				
<20 ^R^	1	[1, 1]	1	[1, 1]
20–24	0.617 ***	[0.484, 0.788]	0.477 ***	[0.340, 0.670]
25–49	0.825	[0.628, 1.085]	0.684 *	[0.512, 0.916]
**Gender**				
Boy ^R^	1	[1, 1]		
Girl	0.761 ***	[0.654, 0.884]		
**Early initiation of BF**				
No ^R^	1	[1, 1]	1	[1, 1]
Yes	0.280 ***	[0.239, 0.330]	0.326 ***	[0.257, 0.414]
**Birth order + birth interval**				
1st birth ^R^	1	[1, 1]		
2nd/3rd birth, <2 years interval	0.673 **	[0.527, 0.860]		
4th or higher birth, <2 years interval	0.775	[0.586, 1.025]		
2nd/3rd birth, ≥2 years interval	0.895	[0.634, 1.265]		
4th or higher birth, ≥2 years interval	1.558 **	[1.141, 2.127]		
**Multiple birth**				
No-singleton ^R^	1	[1, 1]	1	[1, 1]
Yes-twins/triplets	6.348 ***	[4.909, 8.209]	5.640 ***	[3.905, 8.146]
**Perceived size at birth**				
Average ^R^	1	[1, 1]	1	[1, 1]
Larger than average	1.056	[0.892, 1.249]	1.378 **	[1.080, 1.757]
Smaller than average	1.647 ***	[1.358, 1.997]	2.099 ***	[1.599, 2.755]
**Desire for pregnancy**				
Wanted later/no more ^R^	1	[1, 1]		
Wanted then	1.289 **	[1.088, 1.525]		
**Number of ANC visits**				
None	1	[1, 1]	1	[1, 1]
1–3 visits	0.794	[0.623, 1.012]	0.795	[0.555, 1.138]
4 or more visits	0.757 *	[0.592, 0.967]	0.669 *	[0.467, 0.959]
**Institutional delivery**				
No ^R^	1	[1, 1]		
Yes	1.264 *	[1.041, 1.534]		
**BMI**				
Underweight ^R^			1	[1, 1]
Healthy			1.528 *	[1.011, 2.309]
Overweight			2.312 ***	[1.425, 3.750]
Obese			3.280 ***	[1.903, 5.652]

* *p* < 0.05, ** *p* < 0.01, *** *p* < 0.001. CI: Confidence Interval; ^R^: Reference category; na: Information not available. Model A: Includes all 19 countries. Model B: Includes 16 countries that had information on mother’s anthropometry.

## Data Availability

The datasets used for the analyses of this article are publicly available in The DHS Program repository. ICF. 2004–2017. Demographic and Health Surveys (various) [Datasets]. Funded by USAID. Rockville, Maryland: ICF [Distributor].

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
