# Peer review of "The Association between Iron and Folic Acid Supplementation and Malaria Prophylaxis and Linear Growth among Children and Neonatal Mortality in Sub-Saharan Africa—A Pooled Analysis"

_nutrients, 2022, doi:10.3390/nu14214496_

Round 1
Reviewer 1 Report
Dear Authors
Greetings
Congrats for the nice work. I send you the attached doc with some suggestions to improve it. Please read it using Adobe. Regards

Author Response
Thank you for these comments and for giving us an opportunity to revise and resubmit the manuscript. We have carefully reviewed the comments and addressed them one by one in detail below. Please see the attachment.

Reviewer 2 Report
The manuscript describes a comprehensive study of the effects of IFA and malaria prophylaxis on a range of neonatal outcomes.
1. Check that all abbreviations are definedin the title, abstract and text body - for example, DHS, HAZ, BMI and VIF.
2. Reword several sentences to state clearly the nature/direction of the associations described (positive or negative) - for example, lines 48-49, 317-330.
3. Line 172-173 - were no records of birth weight available?
4. Data are plural - check, for example line 198.
5. Figures 1 and 2 - add the legends beneath the graphs and consider using a single stack format so that the data total to 100% (rather than 2 or 3 columns).
6. Table 3 - specify where the statistical differences lie when more than 2 groups are compared.
Author Response

(The authors gave the same response as above.)
